# Salt-Induced Membrane-Bound Conformation of the NAC Domain of α-Synuclein Leads to Structural Polymorphism of Amyloid Fibrils

**DOI:** 10.3390/biom15040506

**Published:** 2025-03-31

**Authors:** Ryota Imaura, Koichi Matsuo

**Affiliations:** 1Graduate School of Advanced Science and Engineering, Hiroshima University, Higashi-Hiroshima 739-8511, Japan; 2Research Institute for Synchrotron Radiation Science, Hiroshima University, Higashi-Hiroshima 739-0046, Japan; 3International Institute for Sustainability with Knotted Chiral Meta Matter (WPI-SKCM^2^), Hiroshima University, Higashi-Hiroshima 739-8526, Japan; 4Research Institute for Semiconductor Engineering, Hiroshima University, Higashi-Hiroshima 739-8527, Japan

**Keywords:** Parkinson’s disease, α-synuclein, non-amyloid-β component, protein aggregation, polymorphism, lipid membrane, membrane interaction mechanism, synchrotron radiation circular dichroism

## Abstract

α-Synuclein (αS) interacts with lipid membranes in neurons to form amyloid fibrils that contribute to Parkinson’s disease, and its non-amyloid-β component domain is critical in the fibrillation. In this study, the salt (NaCl) effect on the membrane interaction and fibril formation of αS_57–102_ peptide (containing the non-amyloid-β component domain) was characterized at the molecular level because the αS_57–102_ fibrils exhibited structural polymorphism with two morphologies (thin and thick) in the presence of NaCl but showed one morphology (thin) in the absence of NaCl. The membrane-bound conformation (before fibrillation) of αS_57–102_ had two helical regions (first and second) on the membrane regardless of salt, but the length of the first region largely shortened when NaCl was present, exposing its hydrophobic area to the solvent. The exposed region induced two distinct pathways of fibril nucleation, depending on the molar ratios of free and membrane-bound αS_57–102_: one from the association of free αS_57–102_ with membrane-bound αS_57–102_ and the other from the assembly among membrane-bound αS_57–102_. The differences mainly affected the β-strand orientation and helical content within the fibril conformations, probably contributing to the thickness degree, leading to structural polymorphism.

## 1. Introduction

Among neurodegenerative diseases, such as Alzheimer’s disease, Parkinson’s disease, and amyotrophic lateral sclerosis [1,2], Parkinson’s disease is induced by the occurrence of Lewy bodies in neurons, which consist of aggregates or amyloid fibrils of α-synuclein (αS) formed on the synaptic membrane [3]. The mechanism of fibril formation of αS has been studied at the molecular level using various spectroscopic methods. These studies revealed that disordered αS at the native state (N state) in solution [4] undergoes a conformational change to an α-helical structure upon interacting with membranes [5]. αS at the membrane-bound (MB) state undergoes a further conformational transition to a β-strand structure to create fibril nucleation and form amyloid fibrils via the continuous abnormal accumulation of αS on the membrane [6]. The parameters of fibrillization kinetics, such as fibril elongation rate constants and lag times, depended on the various factors of the membrane, including its constituents (types of lipid molecules), the net charge of head groups, sizes, and shapes (such as micelles and liposomes) [7,8,9], largely affecting their fibril conformations and toxicities [10]. Furthermore, salts such as NaCl and CaCl_2_, which are ubiquitously present in brain cells (with Na^+^ concentrations of 20–150 mM [11], Cl^−^ approximately 100 mM [12], and Ca^2+^ in the micromolar range [13]), were also important modulators of the parameters of fibrillization kinetics because the salts largely affected the intramolecular and intermolecular interactions among αS molecules in solution and/or on membranes [14]. These factors often result in the different morphologies and lead to structural polymorphism of fibrils [15,16], which exhibit varying levels of toxicity in neuronal cells [17]. For example, αS forms paired filaments, whereas in dementia with Lewy bodies, it assembles into bundled multiple filaments, and in multiple system atrophy, it adopts twisted filaments that cannot be divided into two filaments, as well as straight filaments. These structural variations should be associated with differences in disease-specific pathological features, highlighting the importance of further research on structural polymorphisms [18]. Although the various factors, such as types of membrane and salt, made it difficult to understand the processes from the N state to the fibrillization of αS, the structure of the MB state of αS, which is assigned as the initial state before converting into the fibrils, has provided useful information because this state is the essential step toward protein misfolding and fibrillation and further includes the hints for unraveling the fibrillization mechanism and the properties of fibrils, such as polymorphism and toxicity [19,20,21].

αS comprises the N-terminal (residues 1–60), non-amyloid component (NAC) (residues 61–95), and C-terminal (residues 96–140) domains. Among them, the NAC domain, which contains many hydrophobic amino acids, exhibited extremely poor solubility in aqueous solutions, but several structural analyses have disclosed the important features of this domain. For example, the G68–A78 segment has been identified as a key contributor to synuclein aggregation [22]. The V71–V82 segment was particularly involved in the fibril formation, and its removal resulted in a complete loss of fibrillization ability [23]. Additionally, amyloid filaments comprising a 35-residue peptide corresponding to the NAC domain exhibited toxicity toward dopaminergic human neuroblastoma SH-SY5Y cells [24]. Furthermore, some studies using full-length αS have demonstrated that intermolecular interactions between NAC domains promoted β-sheet formation and played a crucial role in nucleation [25]. Hence, the NAC domain is known as the core region responsible for the β-sheet propensity and aggregation or fibrillation of αS [26,27]. However, regarding the structural alternation of this domain on membranes, only Bisaglia et al. characterized the conformation of the NAC domain in the presence of sodium dodecyl sulfate micelles, suggesting that the micelle-bound NAC domain may inhibit the transition from an α-helical to a β-sheet structure, as the residues responsible for the aggregation process are shielded by the micelle [28], and there is limited information available. Hence, further characterizing the pathways leading to the fibril formation of this domain on the membrane would be important to deepen the understanding of the formation mechanisms of amyloid fibrils in full-length αS.

Circular dichroism (CD) spectroscopy is a valuable technique for studying the structure and function of proteins and peptides under physiological conditions, including in the presence of membranes and salts that mimic the environments inside the cell [14,29,30]. Furthermore, the usage of synchrotron radiation (SR) as a light source made it possible to extend the wavelength of CD measurements until the vacuum-ultraviolet (VUV) region, allowing for high-precision determination of contents, numbers of segments, and sequence of secondary structures of MB proteins and peptides [31,32,33]. Additionally, linear dichroism (LD) spectroscopy under the flow condition or high shear strength clearly showed the orientations of helical and strand structures of proteins and peptides on the membrane surface [34]. Furthermore, attenuated total reflection Fourier transform infrared (ATR-FTIR) spectroscopy, which is less affected by light scattering from insoluble aggregates such as amyloid fibrils, can provide simultaneous information on secondary structure contents of MB proteins and peptides and their orientation against the membrane surface [35]. Although these spectroscopic techniques are limited to providing molecular-level information, molecular dynamics (MD) simulations, based on the secondary structure data obtained from the experimental techniques, enable a deeper understanding of the membrane interaction mechanisms of proteins and peptides at the amino acid level [32]. Hence, combining SRCD spectroscopy [36,37], LD spectroscopy [38], ATR-FTIR, and MD simulations would be a powerful tool for characterizing the MB conformations and fibril structures of the NAC domain and contribute to disclosing the unique fibrillization mechanisms, which are largely related to properties of fibrils such as the polymorphism and toxicity degree.

In this study, we used a variant of the αS_57–102_ fragment [28,39] with multiple charged residues at both ends of the NAC region to improve solubility and to investigate the effect of salt (NaCl) on the amyloid fibril formation of αS_57–102_ under the membrane (50 nm diameter, neutral or anionic membranes, which are a suitable model membrane that mimics the components of synaptic membranes) using thioflavin T (ThT) fluorescence and transmission electron microscopy (TEM). The conformation of membrane-bound αS_57–102_ (precursor to fibril formation) was further characterized using SRCD and LD spectroscopy in the presence or absence of NaCl, and based on these experimental data, the mechanisms of membrane interactions were clarified at the secondary structure and amino acid residue levels using MD simulations. Finally, the effect of salt on the fibril conformation was revealed using ATR-FTIR spectroscopy based on the orientations of β-strand structures in the fibrils. Through the comprehensive analysis of the effects of salts on αS_57–102_ membrane interactions and fibrillization, we provide new insights into the pathways from the MB state to fibril formation and the crucial factors contributing to fibril polymorphism.

## 2. Materials and Methods

### 2.1. Materials

The αS peptide (${\alpha S}_{57–102})$ was synthesized by GL Biochem (Shanghai, China), and the purification and molecular weight were confirmed using high-performance liquid chromatography (>95%) and mass spectroscopy, respectively. The ${\alpha S}_{57–102}$ fragment was chosen because it contains the NAC region, which is established as the core domain responsible for the β-sheet propensity and fibrillation of α-synuclein. 1,2-dimyristoyl-sn-glycero-3-phosphoglycerol (DMPG) and 1,2-dimyristoyl-*sn*-glycero-3-phosphocholine (DMPC) were obtained from Cayman Chemical (Ann Arbor, MI, USA) and Avanti Polar Lipids, Inc. (Alabaster, AL, USA), respectively.

### 2.2. Sample Preparation

Lipid membranes (composed of DMPG or DMPC) with a diameter of 50 nm were prepared using an extrusion technique [40]. DMPG and DMPC, which are model negative and neutral lipid molecules, respectively, were used to investigate the effect of membrane surface charge on α-syn fibrillization since negative and neutral membranes are found in synaptic vesicle membranes [41]. With phase transition temperatures around 23–24 °C [42], both lipids remain in the liquid crystalline phase at experimental temperatures (room temperature or 37 °C), ensuring stability and reproducibility. These features make DMPG and DMPC ideal for studying membrane-dependent α-synuclein aggregation under controlled conditions. Briefly, lipid molecules were dissolved in 20 mM phosphate buffer (pH 6.8), and the freeze–thaw cycles were repeated at least five times by cooling with liquid nitrogen and heating with a thermostatic bath. The lipid solution was then passed 25 times through a 50 nm polycarbonate membrane (Whatman, NJ, USA) using a Mini-Extruder (Avanti, AL, USA). The obtained lipid membranes were subsequently mixed with an ${\alpha S}_{57–102}$ solution as a final ${\alpha S}_{57-102}$ concentration became 50 µM. The concentration of ${\alpha S}_{57–102}$ was determined using the absorbance at 205 nm [43], using the molar extinction coefficient of the ${\alpha S}_{57–102}$ (ε = 135,300 $M^{-1}{\mathrm{cm}}^{-1}$). The same samples were also prepared in the presence of 0.1 M NaCl to mimic the salt condition inside brain cells [11].

### 2.3. ThT Fluorescence Assay

Fluorescence assays of ${\alpha S}_{57–102}$ in the presence or absence of NaCl were conducted using a fluorescence spectrophotometer (FP-3800, JASCO, Tokyo, Japan), with ThT employed as the amyloid-specific fluorescent dye. The concentration of ${\alpha S}_{57–102}$ was fixed at 50 µM, and lipid vesicles were prepared at L/P molar ratios of 0, 20, and 100. An L/P ratio of 0 corresponds to a control condition without a lipid membrane. ThT was prepared at a final concentration of 10 µM. Amyloid fibril formation was initiated by incubating the samples at 1500 rpm and 37 °C in a block bath shaker (HCM100-Pro, DLAB, Beijing, China). Fluorescence emission spectra were recorded with an excitation wavelength of 450 nm and an emission wavelength of 480 nm. The ThT fluorescence spectra of each sample were measured at 1 h intervals over a 15-h period. Each measurement was repeated three times and averaged. The same experiments were also repeated at least twice. The baselines, which were obtained from the measurements of ThT fluorescence in the absence of the peptide (i.e., only lipid membranes composed of DMPG or DMPC), did not show noticeable intensity as a function of incubation time as shown in Appendix A. The kinetics of ${\alpha S}_{57–102}$ aggregations were monitored by plotting ThT fluorescence intensity as a function of time, and the data were fitted to a sigmoidal growth model using the following equation [44]:$$Y=y_{I}+m_{I}x+\frac{y_{F}+m_{F}x}{1+e^{-\left[k\left(x-x_{0}\right)\right]}}$$
where Y denotes the fluorescence intensity, x is the time, and $y_{I}+m_{I}x$ and $y_{F}+m_{F}x$ represent the baseline drift in the initial (I) and final (F) phases, respectively, with y and m as the intercept and slope. The parameter $x_{0}$ represents the midpoint or the time required to reach 50% of the maximum fluorescence intensity, while k defines the apparent rate constant of fibril elongation.

### 2.4. TEM

High-resolution TEM images were acquired using a JEOL JEM-2100 microscope (JEOL, Tokyo, Japan) operated at an accelerating voltage of 200 kV at the N-BARD, Hiroshima University. Samples for TEM imaging were prepared by incubating the ${\alpha S}_{57–102}$ at 1500 rpm and 37 °C for 10 h using the block bath shaker. The prepared samples were adsorbed onto carbon-coated copper grids, rinsed with water, and then negatively stained with 2% (*w*/*v*) uranyl acetate [45].

### 2.5. SRCD Measurement

The SRCD spectra of the N and MB states of ${\alpha S}_{57–102}$ in the presence or absence of NaCl were collected using the SRCD spectrophotometer at beamline BL12 of the Research Institute for Synchrotron Radiation Science, Hiroshima University [30,46]. The concentration of ${\alpha S}_{57–102}$ was kept at 50 µM, and lipid membranes were prepared at L/P molar ratios ranging from 0 to 200 in the absence of NaCl and 0 to 300 in the presence of NaCl. All SRCD measurements were conducted at 25 °C, with an optical path length of 100 µm, a scanning speed of 20 nm/min, and averaged over eight accumulations in the wavelength range of 260–175 nm. All SRCD spectra were constant within an experimental error of 5% during data acquisition, which was estimated by six accumulations. To minimize light scattering from the liposomal particles, the optical cell was positioned within 10 mm of the photomultiplier tube detector [29,47]. The baseline for SRCD spectra was obtained using samples containing lipid membranes at the corresponding concentrations without the ${\alpha S}_{57–102}$.

### 2.6. Secondary Structure Analysis

The contents and numbers of segments of secondary structures of ${\alpha S}_{57–102}$ under each experimental condition of L/P ratios were analyzed using the corresponding SRCD spectra and the SELCON3 PROGRAM, in which a database composed of SRCD spectra and secondary structure content data of 31 reference proteins is used [48]. Additionally, these secondary structure data obtained from SRCD spectra were used to predict the positions of α-helices and β-strands based on a neural network method (VUVCD-NN) [49]. This prediction used a training dataset of X-ray structural data of 607 proteins retrieved from the PDB to calculate the weights and biases for α-helices and β-strands for each amino acid. Predictions were repeated until the secondary structure parameters (contents and numbers of segments) converged with those estimated by SRCD spectra. The accuracy of this approach was 74.9% for 30 soluble proteins and 73% for 15 transmembrane proteins [32]. These analytical methods have been used to estimate the secondary structures of membrane proteins [50] and various MB proteins, disclosing the details of structural alternations from the N to MB states [31,32,33,34].

### 2.7. Kinetics Analysis

A two-state transition model was employed to calculate the membrane binding ratio and dissociation constant $K_{d}$ of ${\alpha S}_{57–102}$ from the CD data [51]. The increase in CD signal at 222 nm as a function of the L/P ratio corresponds to the enhancement of α-helical content resulting from membrane binding. The observed CD signal ($CD_{obs}$) can be described by the following equation:$$CD_{obs}=CD_{\alpha S_{free}}\cdot X_{\alpha S_{free}}+CD_{\alpha S_{bound}}\cdot X_{\alpha S_{bound}}$$

$X_{\alpha S_{free}}$ and $X_{\alpha S_{bound}}$ represent the fractions of free and membrane-bound ${\alpha S}_{57–102}$, respectively, with the relationship $X_{\alpha S_{free}}+X_{\alpha S_{bound}}$ = 1. $CD_{\alpha S_{free}}$, and $CD_{\alpha S_{bound}}$ correspond to the CD intensities of ${\alpha S}_{57–102}$ in the N and MB states, respectively. The dissociation constant ($K_{d}$) was determined under the assumption that free ${\alpha S}_{57–102}$ interacts with multiple phospholipid molecules in the membrane, based on the following equation:(1)$$\alpha S_{free}+Lipid_{N}\overset{K_{d}}{↔}\;\alpha S_{bound}:Lipid_{N}$$(2)$$K_{d}=\frac{\left[\alpha S_{free}\right]\left[Lipid_{N}\right]}{\left[\alpha S_{bound}:Lipid_{N}\right]}$$

The concentrations of free ${\alpha S}_{57–102}$ and lipid molecules available for binding to ${\alpha S}_{57–102}$ are denoted as $\left[\alpha S_{free}\right]$ and $\left[Lipid_{N}\right]$, respectively. The concentration of ${\alpha S}_{57–102}$ complexes is represented as $\left[\alpha S_{bound}:Lipid_{N}\right]$. The total concentrations of ${\alpha S}_{57–102}$
${\left[\alpha S\right]}_{T}$ and lipid ${\left[Lipid\right]}_{T}$ are defined as follows:(3)$${\left[\alpha S\right]}_{T}=[\alpha S_{free}]+[\alpha S_{bound}\;:\;Lipid_{N}]$$(4)$${\left[Lipid\right]}_{T}=N\left([Lipid_{N}\right]+[\alpha S_{bound}\;:\;Lipid_{N}])$$

From (2), (3), and (4),$$[\alpha S_{bound}\;:\;Lipid_{N}]=\frac{({\left[\alpha S\right]}_{T}+\frac{{\left[Lipid\right]}_{T}}{N}+K_{d})-\sqrt{{\left({\left[\alpha S\right]}_{T}+\frac{{\left[Lipid\right]}_{T}}{N}+K_{d}\right)}^{2}-\frac{4({\left[\alpha S\right]}_{T}{\left[Lipid\right]}_{T}}{N}}}{2}$$

From the above equations, the following expression for $X_{\alpha S_{bound}}$ can be obtained:(5)$$X_{\alpha S_{bound}}=\frac{({\left[\alpha S\right]}_{T}+\frac{{\left[Lipid\right]}_{T}}{N}+K_{d})-\sqrt{{\left({\left[\alpha S\right]}_{T}+\frac{{\left[Lipid\right]}_{T}}{N}+K_{d}\right)}^{2}-\frac{4({\left[\alpha S\right]}_{T}{\left[Lipid\right]}_{T}}{N}}}{2{\left[\alpha S\right]}_{T}}$$

Fitting of Equations (1) and (5) was conducted using the least squares method, with $K_{d}$ and *N* treated as adjustable parameters [51].

### 2.8. LD Measurement

LD measurements of the MB state of ${\alpha S}_{57–102}$, in the presence or absence of NaCl, were conducted using a Couette flow cell, which was set up into a CD spectrophotometer (J-1500, JASCO, Japan) [52]. The Couette flow cell consists of a cylindrical chamber with a rod at the center, where the sample continuously flows between two concentric cylinders—one rotating and the other stationary. The sample was confined between the cylinder and the rod, and shear flow was generated in the narrow gap by rotating the rod, inducing deformation of the liposomes from spherical to elliptical shapes. This deformation induces the positive or negative LD signal, originated from the average orientation of helical structures on the membrane surface (helical axis is parallel or perpendicular to the membrane surface) [53,54]. The concentration of αS was 50 µM, and lipid membranes were prepared as the L/P ratio became 200. LD spectra were recorded at 25 °C with an optical path length of 0.5 mm, a data acquisition interval of 1 nm, a scanning speed of 20 nm/min, and averaged over two accumulations in the wavelength range of 350 to 200 nm. The baselines were obtained from the spectra of each sample without flow conditions.

### 2.9. Data Analysis for LD Spectra

The effect of light scattering on the LD spectrum can be estimated as follows [52]:$$LD^{obs}=LD^{sample}+LD^{scattering}$$

$LD^{obs}$, $LD^{sample}$, and $LD^{scattering}$ represent the experimentally observed LD, LD from the sample, and LD contribution from light scattering, respectively. The $LD^{scattering}$ component was determined using an empirical correction fitting equation proposed by Nordén et al. and was estimated from the LD spectrum in the wavelength range of 300–350 nm. The equation used to model the light scattering contribution is:$$LD^{scattering}=a\lambda^{-k}$$
where a is a flexible parameter, λ is the wavelength, and k is the constant parameter depending on the sample solutions, which range from 2.8 to 3.5.

### 2.10. ATR-FTIR Measurement

The FTIR absorbance spectra of membrane-bound and fibrillated ${\alpha S}_{57–102}$ were measured in the presence or absence of NaCl using an ATR-FTIR spectrometer (Jasco FT/IR 4600 with ATR PRO610P-S) equipped with a ZnSe prism to determine the orientation of α-helical and β-sheet structures on the membrane surface [35,55]. The spectra were acquired with a resolution of 4 cm^−1^ and accumulated with 256 scans. The measurements were repeated at least twice. Samples were prepared under the same conditions as for the SRCD and LD measurements, deposited onto SiO_2_ windows, and dried under vacuum for 24 h. The baselines were obtained from the spectra of the SiO_2_ crystal without samples (lipid and peptide). The absorbance (A) of the amide I band (1700–1600 cm^−1^) from the polarized ATR-FTIR was obtained from the following formula:$$A=A_{∥}+GA_{⊥}$$

$A_{∥}$ and $A_{⊥}$ are the absorbances of the amide I band for the irradiation polarized parallel and perpendicular, respectively, to the plane of incidence of the infrared beam. $G\;\left(=2\frac{E_{z}^{2}}{E_{y}^{2}}-\frac{E_{x}^{2}}{E_{y}^{2}}=2.65,\;\frac{E_{x}^{2}}{E_{y}^{2}}=0.450,\;\frac{E_{z}^{2}}{E_{y}^{2}}=1.550\right)$ is a constant parameter of the thick film on the ZnSe crystal, which was given by a previous paper [56]. Band width and position of each peak were determined from the second derivative of absorbance spectra, and Gaussian fitting [57] was performed using these parameters to optimize the peak height. Subsequently, both the width (FWHH; 5–30 cm^−1^ [58]) and position of the peaks were varied simultaneously, and the fitting procedure was iterated until the coefficient of determination (R^2^) between experimental and fitted spectra exceeded 0.996.

The absorbances of amide I bands were also used to infer the orientations of the α-helical and β-sheet structures relative to the membrane surface. The α-helix component ratio ($R^{ATR\_\alpha }$) is calculated using the following formula:$$R^{ATR\_\alpha }=\frac{R^{aI}-\frac{R^{aI}+2}{2R^{iso}+1}\left(1-x\right)}{1-\frac{1}{R^{iso}}\frac{R^{aI}+2}{2R^{iso}+1}\left(1-x\right)}$$

$R^{aI}$ represents the dichroic ratio, which is determined by the ratio of the $A_{∥}$ and $A_{⊥}$ absorbances of α-helix components in the amide I band ($R^{aI}$ = $A_{∥}/A_{⊥}$), *x* denotes the fraction of the α-helix component, and $R^{iso}$ is the dichroic ratio of the lipid carbonyl band (1740 cm^−1^). $R^{ATR\_\alpha }$ relates to an experimental order parameter ($S_{exp}$) as follows:$$R^{ATR\_\alpha }=\frac{E_{z}^{2}}{E_{y}^{2}}+\frac{E_{x}^{2}}{E_{y}^{2}}\left(1+\frac{3S_{exp}}{1-S_{exp}}\right)$$

From this equation, $S_{exp}$ can be derived, and further, from $S_{exp}$, $\theta_{helix}$, which is the angle between the α-helix axis and the membrane normal, can be obtained from the following two equations:$$S_{exp}=S_{ch}S_{helix}S_{dip}$$$$S_{helix}=\frac{1}{2}\left(3\;\cos^{2}\theta_{helix}-1\right)$$

$S_{ch}$ is the order parameter related to the CH_2_ stretching vibration bands at both 2850 cm^−1^ (symmetric) and 2920 cm^−1^ (asymmetric) in the lipid molecules, and $S_{\mathrm{dip}}$ represents the orientation of the amide I dipole (α-helix), which is oriented between 27° and 36°.

The orientations of the β-sheet and the tilt angle of the strands are calculated from the dichroic ratios of the amide I and amide II bands ($R^{\beta }$= $A_{∥}/A_{⊥}$) which are determined by the ratio of the $A_{∥}$ and $A_{⊥}$ absorbances of β-sheet components in the amide I band and amide II bands, respectively, as shown in the following equations:$$R^{\beta I}=\frac{E_{x}^{2}}{E_{y}^{2}}+\frac{2\langle \cos^{2}\alpha \rangle \langle \sin^{2}\beta \rangle }{1-\langle \cos^{2}\alpha \rangle \langle \sin^{2}\beta \rangle }\frac{E_{z}^{2}}{E_{y}^{2}}$$$$R^{\beta II}=\frac{E_{x}^{2}}{E_{y}^{2}}+\frac{2\langle \cos^{2}\alpha \rangle \langle \cos^{2}\beta \rangle }{1-\langle \cos^{2}\alpha \rangle \langle \cos^{2}\beta \rangle }\frac{E_{z}^{2}}{E_{y}^{2}}$$
where α is the angle that β-sheet makes with the membrane normal, and β is the β-strand tilt.

### 2.11. MD Simulation

The initial structure of the E57–K102 fragment of αS (${\alpha S}_{57-102}$) was generated using CHARMM-GUI, based on the Protein Data Bank (PDB) structure of αS (PDB ID: 1XQ8) [59], and a 100 ns MD simulation for this fragment was conducted in aqueous solution at 350 K. The obtained unordered structure was used as the starting conformation for subsequent simulations. The membrane model, consisting of DMPG lipid molecules, was constructed using the highly mobile membrane-mimetic model. To enhance structural sampling of membrane-bound ${\alpha S}_{57–102}$ and account for lipid packing defects, a scaling factor of 1.3 was applied [60]. The peptide fragments were initially positioned 15 Å away from the membrane surface to reduce the effects of strong initial interactions with the membrane [61]. The NaCl concentration was adjusted to match the experimental conditions, and the number of phospholipids was set to 200 to mimic the experimental L/P ratio.

MD simulations were performed using the GROMACS software package with the CHARMM36m force field, which is appropriate for both disordered proteins and biological membranes [62,63]. Simulations were run for 500 ns with a 2 fs time step because previous studies have demonstrated that membrane binding of α-syn typically occurs within 100 ns [64] and have suggested that simulations of at least 500 ns are required to effectively observe lipid-protein interactions and their dynamic behavior over time [65]. This allows us to obtain sufficient time for membrane-binding events and capture the subsequent lipid-protein interactions. The temperature was controlled at 298.15 K using the Nose-Hoover thermostat, and pressure was maintained at 1 bar using the NPAT ensemble and the Langevin barostat [66,67]. Periodic boundary conditions were employed throughout the simulations. Electrostatic interactions were calculated using the particle-mesh Ewald method with a 12 Å cutoff, while van der Waals forces were computed using the Lennard-Jones potential with the same cutoff [68]. Covalent bonds involving hydrogen atoms were constrained using the LINCS algorithm [69]. Each MD simulation was performed at least twice, yielding consistent results.

## 3. Results and Discussion

### 3.1. NaCl Induced the Structural Polymorphism in αS_57–102_ Amyloid Fibrils on Anionic Lipid Membranes

The amyloid fibril formations of αS_57–102_ in the N state at lipid-to-peptide (L/P) = 0 (without membrane) and the MB states (anionic DMPG or neutral DMPC lipid membranes) at L/P = 20 and 100 were monitored in the presence or absence of NaCl using ThT fluorescence, as shown in Figure 1 and Appendix A.

In the case of anionic membranes, regardless of NaCl, the ThT fluorescence intensity was not observed at L/P = 0 but was observed at L/P = 20, exhibiting a sigmoidal curve and a maximum value after approximately 5–6 h, as shown in Figure 1. Furthermore, the intensity slightly occurred at L/P = 100 in the presence of NaCl but not in the absence of NaCl. However, no significant ThT fluorescence was detected at any L/P ratios in the neutral lipid membranes, as shown in Appendix A, indicating that the presence of anionic membranes is an essential factor for amyloid fibril formation by αS_57–102_.

The lag time and elongation rate constant for fibril formation at L/P = 20 in the DMPG lipid membrane were 4.62 h and 3.65 $h^{-1}$ in the presence of NaCl, respectively, which were much slower and comparable to those in the absence of NaCl (1.26 h for lag time and 3.5 $h^{-1}$ for elongation rate constant), which means that the salt mainly contributed to slower nucleation of fibrils. According to the previous findings [70], higher residue flexibility promotes intermolecular interactions between monomers during the early stages of nucleation, thereby accelerating the nucleation process. This suggests that the nucleation delay observed in the presence of NaCl may be attributed to reduced protein flexibility. The finding suggests that ionic conditions, such as the presence of NaCl, might modulate the nucleation of fibrils, potentially affecting the polymorphism of fibrils. The morphology of the fibrils was further examined using TEM, as shown in Figure 2.

In the absence of NaCl, the fibrils at L/P = 20 were clearly thin (Figure 2A), while in the presence of NaCl, two distinct fibril types were identified at L/P = 20: thin fibrils (Figure 2B), and thick fibrils (Figure 2C). Furthermore, at an L/P = 100 in the presence of NaCl, thick fibrils (width = 191.5 ± 19.5 nm) with twist were mainly detected (Appendix A), which was a relatively similar thickness observed at L/P = 20 (Figure 2C). These findings suggested that NaCl induced the structural polymorphism for αS_57–102_ amyloid fibrils on anionic lipid membranes. To further understand the occurrence of unique fibril morphologies depending on NaCl and L/P ratios, the conformations of the MB state, which is considered the initial structure of fibril formations, were investigated using SRCD spectroscopy in the presence or absence of NaCl at various L/P ratios.

### 3.2. NaCl Depressed the Amount and Length of Helical Structures of αS_57–102_ on Membranes

As shown in Figure 3, the SRCD spectra of αS_57–102_ interacted with the DMPG membrane and were measured in the presence or absence of NaCl at various L/P ratios. Regardless of NaCl, the SRCD spectra at the N state (L/P = 0) exhibited a characteristic profile of a random coil structure, with a distinctive negative shoulder of approximately 220 nm and a negative peak at 200 nm, which is consistent with a previous report [28]. In the absence of NaCl, as the L/P ratio increased, the SRCD exhibited a spectral shape of α-helix structure, with negative peaks at 222 and 208 nm and a positive peak at approximately 190 nm. The CD plots at 222 nm (inset of Figure 3A) showed that the intensity saturated at L/P = 50~100, indicating the formation of the MB state. In the presence of NaCl, as the L/P ratio increased, the characteristic peaks of the α-helix structure also appeared, and the CD plots at 222 nm also saturated at approximately L/P = 50~100 (inset of Figure 3B), as observed in the case of the absence of NaCl. However, the CD intensity in the MB state was much lower than that observed in the absence of NaCl (Figure 3). Additionally, in the absence and presence of NaCl, an iso-ellipticity point was observed at approximately 204 and 203 nm, respectively, suggesting that αS_57–102_ had two-state transitions from the N to MB states without any intermediates. However, the CD spectra showed no structural changes in the DMPC lipid membrane, regardless of the presence of salt (Appendix A), and further fibril formation was not observed (Appendix A), showing that the structural alternation of αS_57–102_ on anionic membranes is crucial for the following fibril formation. These results were supported by several studies in which the negative charge on the synaptic membranes is essential for the interaction with αS [71,72]. However, the differences in the iso-ellipticity point and CD intensity implied that the respective MB state in the presence and absence of NaCl had different conformations. To further distinguish the differences, the contents and the number of segments of secondary structures were analyzed using the spectra at L/P = 200 [73,74], and the results are listed in Table 1.

In the L/P = 0 (N state), the α-helix content was 0% and 2.3%, and the extended β-strand content [75] was 36.2% and 29.1% in the absence and presence of NaCl, respectively. Although the N states differ slightly under both conditions, this difference would arise from intramolecular structural changes as observed in full-length αS [76]. Upon membrane binding, the α-helix content of αS_57–102_ significantly increased to 66.4% in the absence of NaCl, while the extended β-strand content markedly decreased. Furthermore, the number of α-helix segments increased from 0 to 2. In the presence of NaCl, the α-helix content and the number of α-helix segments were 37.6% and 2, respectively. Compared with the results in the absence of NaCl, the α-helix content decreased by 28.8% (equivalent to approximately 13 residues), whereas the extended β-strand content increased by 11.7% (approximately five residues). The salt-induced reduction in α-helix content is consistent with the behavior observed in our previous study for the full-length of αS [14]. The reduction in α-helix content in the presence of NaCl suggests that the non-helical region of the membrane-bound αS_57–102_ might be exposed to the solvent to promote the transition to amyloid fibrils owing to the intermolecular interaction with free αS_57–102_, inducing the structural polymorphism of the fibrils. Since CD spectra do not provide direct information on the specific location of secondary structures at the amino acid sequence level, the VUVCD-NN analysis was conducted to predict the positions of secondary structures at N and MB states by combining the experimentally obtained secondary structure data (contents and number of segments) and the amino acid sequence of the αS_57–102_ fragment [31,32,33,49,77,78]. The predicted positions of the α-helical regions in the MB state are shown in Figure 4.

As shown in Figure 4, in the presence of NaCl, the α-helix regions of αS were assigned to G68–G84 and I88–L100, whereas in the absence of NaCl, the regions were predicted to be V74–K80 and A90–Q99. The number of α-helical regions was commonly two (first and second helical regions) in both states, but their length was shorter in the presence of NaCl, in particular for the first helical region. These findings suggest that NaCl depressed the amount and length of the helical structure of αS_57–102_ on the membrane, which might contribute to the structural polymorphism in amyloid fibrils, as observed in Figure 2B,C. These predicted results were supported by previous papers that reported that the area around I88–L100 was closely related to the membrane-penetrating region [79] and that the area around G68–A76 was involved in fibril formation [23]. Additionally, the positions of the helical regions in the αS_57–102_ were predicted using AlphaFold2, JPRED, PSIPRED, and SOPMA (Appendix A). These four methods presented much different results from the experimentally determined contents and numbers of segments of the helix (Table 1), highlighting the effectiveness of our approach for predicting the positions of secondary structures of MB proteins as introduced in several reports in which the VUVCD-NN method was applied to characterize the conformation of MB proteins and peptides [31,32,33,49,77,78].

### 3.3. NaCl Did Not Affect the Average Orientation of the Helical Structure on the Membrane Surface

Figure 5 shows the LD spectra of the MB state of αS_57–102_ in the absence or presence of NaCl. No LD signal was observed for the N state. A positive peak was detected at approximately 200 nm at L/P = 200, and its intensity increased as the rotation speed increased. The LD signals below 250 nm should be assigned as the orientation of α-helical structures, where the electric dipole moments are oriented perpendicular to the helical axis at 190 nm, parallel at 208 nm, and perpendicular at 222 nm [52], and hence these LD signals proved that the helical regions of αS directly interact with the surface of DMPG lipid membranes, regardless of NaCl. In the LD signal, it is expected that the contributions of β-strands would be negligible or small because the α-helix content is at least three times higher than that of β-strands, and the LD signal showed a positive correlation with the amount of α-helical structure (Table 1).

The inset shows the results of the fitting analysis for the spectra at 3000 rpm, which assumes that all α-helical structures of αS_57–102_ align uniformly at a single angle relative to the membrane surface [34]. The fitting results reproduced the experimental data well. In this assumption, the orientation angles of the α-helical regions of αS_57–102_ on the DMPG lipid membrane were calculated to be 46.7± 0.1° in the absence of NaCl, and 47.3±0.2° in the presence of NaCl, showing no significant difference between the two conditions. Additionally, ATR-FTIR spectroscopy was employed to investigate the orientations of helical structures of αS_57–102_ [35,80]. This method is also a valuable technique for observing the MB conformations of proteins and peptides that are difficult to analyze using X-ray crystallography or nuclear magnetic resonance. The results for αS_57–102_ revealed that, in the absence and presence of NaCl, it is oriented at approximately 50° relative to the membrane normal, which value matched well the angle calculated from LD. These findings suggest that the helical orientation against the membrane surface was similar regardless of NaCl and that the structural polymorphism of the fibrils might not be strongly attributed to the helical orientations on the membrane, as shown in Figure 2.

### 3.4. NaCl Significantly Exposed the First Helical Regions of αS_57–102_ to the Solvent

SRCD and LD analyses revealed that αS_57–102_ can interact with DMPG lipid membranes regardless of NaCl, although the number of interaction regions was reduced in the presence of NaCl. However, these experimental methods cannot identify the key amino acid residues involved in the interaction with the membrane. To address these issues, MD simulations, which can provide the structural information of proteins and peptides at the amino acid residue level, were performed for αS_57–102_ on the membranes for 500 ns. In this study, the αS_57–102_ fragment with a disordered structure was initially positioned 15 Å away from the membrane surface (see Section 2) and then simulated. Owing to the limited simulation time, the formation of secondary structures was not observed, but over longer timescales (beyond the microsecond scale), using coarse-grained simulations or high sampling techniques to reduce computational costs [81] would allow us to see the conformational changes. Hence, the results obtained here should reflect the early stages of membrane interactions of αS_57–102_.

The root-mean-square fluctuation (RMSF) of each amino acid residue was first extracted from the results of MD simulations (Appendix A), and it was shown that NaCl totally reduced the residue fluctuations across the fragment. As mentioned above, higher residue flexibility generally facilitates the intermolecular interactions between monomers during the early stages of nucleation, accelerating the nucleation process [70]. Hence, the RMSF from the MD simulation supported our experimental result in which the lag time (the nucleation time) was faster in the absence of NaCl (Figure 1). Further, the reduction in RMSF or low flexibility due to the presence of NaCl might stabilize the conformation of membrane-bound αS_57–102_, potentially facilitating an alternative nucleation pathway, and the type of pathway in the presence of NaCl would be affected by the L/P ratios, as expected from Figure 2 and Appendix A

Figure 6A shows the time dependence of the distance between the center of gravity of αS_57–102_ against the position of the phosphorus atom in DMPG lipid molecules. During the simulations, the center of mass of the αS_57–102_ approached the membrane surface at approximately 150 ns in the absence of NaCl and approximately 60 ns in the presence of NaCl, and both fragments stayed near the membrane surface except for some dumps at approximately 100 ns in the presence of NaCl (Figure 6A). The average positions of each amino acid residue between 490 and 500 ns were further analyzed, as shown in Figure 6B, and revealed that the N65–V77 and I88–V95 regions of αS_57–102_ directly interacted with the membrane in the absence of NaCl, while the regions V74–Q79 and I88–F94 directly interacted with the membrane in the presence of NaCl (Figure 6B). The membrane-interacted regions from the simulations mostly identified with the α-helix regions predicted using the VUVCD-NN method (G68-G84 and I88-L100 for the absence of NaCl and V74-K80 and A90-Q99 for the presence of NaCl), indicating the usefulness of combining the CD experiments and MD simulations for the characterization of αS_57–102_ conformation on the membrane. From these simulations, we can also estimate that the decreased helical contents in the presence of NaCl (Table 1) would be attributed to the decrements in the membrane interaction regions of αS_57–102_.

Although the regions of membrane penetration were reduced in the presence of NaCl, probably due to the reduction in electrostatic effects, the membrane interactions of the segments were commonly mediated by hydrophobic interactions because the N65–V77 segment (total: 13 residues) and the I88–V95 segment (total: eight residues) in the absence of NaCl contain seven (A and V) and six (I, A, F, and V) hydrophobic residues, respectively, and the V74–Q79 segment (total: six residues) and the I88–F94 segment (total: seven residues) in the presence of NaCl contain four (A and V) and six (I, A, F, and V) hydrophobic residues, respectively. Furthermore, the second helical regions of both conditions (I88–V95 and I88–F94 for absence and presence of NaCl, respectively) commonly interacted with the membrane (Figure 6B) and deeply penetrated into the membrane (Appendix A), which suggests that the membrane interaction of the αS_57–102_ fragment would be primarily driven by the hydrophobic interactions of the second helical region. This suggestion was supported by Shvadchak et al., who reported that the area, including the second region observed in this study (around I88–L100), was closely related to the membrane-penetrating region [79]. However, the first helical region largely shortened in the presence of NaCl (N65–V77 and V74–Q79 for the absence and presence of NaCl, respectively), and the difference area (mainly N65-G73) was exposed to the solvent (Figure 6B). Previous studies have also demonstrated that the area corresponding to the first region was involved in fibril formation. For example, replacing A69–V75 with KTKEGV (which is the repeat motif of the N-terminal region of αS) inhibited fibril formation [82], and the G68–A76 segment has been identified as critical for β-sheet formation and toxicity [23]. Although the above two studies targeted the full-length αS, these findings suggested that the exposed area might be a key factor in inducing the fibril formation and structural polymorphism, as shown in Figure 1 and Figure 2.

### 3.5. Structural Polymorphism of αS_57–102_ Fibrils Occurred Owing to Different Fibrillation Pathways

As mentioned above, we found that the amount of the two morphologies depended on the L/P ratios in the presence of NaCl (the amount of thin fibrils was larger at L/P = 20, but the amount of thick ones was larger at L/P = 100), while only one morphology was observed in the absence of NaCl. The different L/P ratios affected the amount of free αS_57–102_ in solution or bound αS_57–102_ in the membrane, and hence the populations of membrane-bound αS_57–102_ at various L/P ratios were estimated using a two-state transition model function, as shown in Figure 7.

From this figure, the population of membrane-bound αS_57–102_ increases with increasing L/P ratios, reaching a plateau at approximately L/P = 80 and 100 in the absence and presence of NaCl, respectively. Furthermore, $K_{d}$ in the absence of NaCl was smaller than in the case of NaCl presence (4.94±0.47 and 8.22±0.04 µM in the absence and presence of NaCl, respectively). This might be probably due to the inhibitions of the electrostatic interactions between αS_57–102_ and the membrane by NaCl. In the absence of NaCl, the fibrils with thin morphology occurred at L/P = 20 (Figure 2A), in which the population of membrane-bound αS_57–102_ was 42%, while the fibrils were not observed at L/P = 100, in which the population of the bound population was 94%, indicating that the abundant free αS_57–102_ at L/P = 20 can directly interact with membrane-bound αS_57–102_ to make the nucleation and facilitate fibril formation, whereas the small amount of free αS_57–102_ at L/P = 100 should be insufficient to make the nucleation via the pathway as observed at L/P = 20. In the presence of NaCl, the fibrils formed at both L/P = 20 and 100, in which the populations of membrane-bound αS_57–102_ were 36% and 89%, respectively (which corresponds to 64% and 11% of free αS_57–102_, respectively). At L/P = 20, the abundant free αS_57–102_ can directly interact with membrane-bound αS_57–102_ to make the nucleation and mainly facilitate thin fibril formation, as observed in the absence of NaCl (Figure 2B), while at L/P = 100, owing to the insufficient free αS_57–102_, rather than the direct interaction between MB and free αS_57–102_, the MB αS_57–102_ gathered together and assembled to make the nucleation and facilitate fibril formation with thick and twisted fibrils (Appendix A). The elongation rate constant was 1.03 $h^{-1}$ (L/P = 100), which is much smaller than 3.65 $h^{-1}$ (L/P = 20), indicating that the slower accumulation of fibrils might provide the thicker morphology.

The differences in the MB conformations of αS_57–102_ at L/P = 100 in the presence and absence of NaCl were mainly addressed in the solvent-exposed area (N65-G73) (Figure 6B), which includes several hydrophobic amino acid residues such as V66, A69, V70, and V71. Furthermore, the simulation of fibril formation of NAC peptides showed that the interactions between peptides occurred when the amino acids around V70 of each peptide approached each other, suggesting that V70 played a crucial role in promoting β-sheet formation in αS [83]. These results indicate that, at L/P = 100, the salt induced the exposure of some hydrophobic amino acids into solvent and promoted the interactions among the hydrophobic areas of αS_57–102_ molecules on the membrane [19], forming the aggregation or nucleation. The accumulation of aggregates would be possible even at L/P = 100, where the average population of membrane-bound αS_57–102_ was 89% because the binding or interaction of free αS_57–102_ with membranes was reversible and in equilibrium.

Based on the observations of two distinct fibril types in the presence of NaCl, two fibril formation pathways are proposed, contributing to fibril polymorphism: (1) the formation of amyloid fibrils started from the nucleation due to the intermolecular interactions of free αS_57–102_ with the membrane-bound αS_57–102_ and the fibrils were elongated by additional interaction of free αS_57–102_ (cases of L/P = 20 in the presence and absence of NaCl) [84], and (2) the fibril formation initiated from the nucleation among some membrane-bound αS_57–102_, followed by the interaction with free αS_57–102_ for the elongation (case of L/P = 100 in the presence of NaCl) [19].

The effect of KCl, which is a similar salt to NaCl, on the structural polymorphism of fibrils was not investigated in our study. However, we expect that KCl does not induce the polymorphism as observed in NaCl presence because Havemeister et al. reported that NaCl accelerated fibril formation while KCl exhibited an inhibitory effect on fibril assembly [85]. Mechanistically, Na^+^ ions effectively neutralize electrostatic repulsion and stabilize β-sheet-rich fibrillar structures, facilitating the nucleation and growth of fibrils and contributing to the formation of multiple polymorphic forms. Conversely, K⁺ ions lead to a weaker reduction in electrostatic repulsion to maintain the monomeric protein in a random coil state, inhibiting the formation of fibrils. Additionally, the fibrils formed in the presence of KCl increased its rigidity and decreased its clustering behavior, which made it difficult to form the fibril polymorphisms as observed in the presence of NaCl [85]. However, this study was conducted without lipid membranes, and hence, further experimental validation under membrane-interacted conditions may be necessary to confirm the salt dependence of fibril polymorphisms.

### 3.6. Structural Differences in Polymorphisms of αS_57–102_ Fibrils Were Originated from the Orientations of β-Strands

The occurrence of structural polymorphism in fibrils is very sensitive to the solvent environments [15,16]. In our study, TEM analyses revealed that the αS_57–102_ forms thin fibrils in the absence of NaCl at an L/P = 20 (Figure 2A), while in the presence of NaCl at an L/P = 20, it majorly forms thin fibrils and minorly thick fibrils (Figure 2B,C), and, at an L/P = 100, it mainly forms thick fibrils with twist (Appendix A), which had a similar thickness to minor fibrils at L/P = 20 (Figure 2C), indicating that NaCl is an essential factor to induce structural polymorphism (Figure 2). To compare between the fibril conformations at the molecular level in the presence and absence of NaCl, we analyzed ATR-FTIR data of fibrils because the data can disclose the orientation angle of β-sheets relative to the membrane normal (angle α) and the tilt angle of β-strands within the β-sheet structure (angle β) (Appendix A) [35], which were important factors to determine the morphologies of amyloid fibrils [41,86,87]. The structural analysis of fibrils based on ATR-FTIR is summarized in Table 2.

ATR-FTIR analysis revealed that fibrils at an L/P = 20 in the presence of NaCl exhibited a higher β-sheet content and a larger angle β than those in the absence of NaCl, while the angle α showed no significant difference between both conditions. In contrast, the fibril formed at an L/P = 100 in the presence of NaCl reduced the β-sheet content and increased and decreased the angles β and α, respectively, compared with those formed at an L/P = 20. It has been reported that the sheet stacking along the protofilament axis related to the twisted morphology was very sensitive to the change in angles β and α within fibrils [86]. Furthermore, the fibrils formed at L/P = 100 showed a higher helical content than fibrils formed at L/P = 20, which is consistent with reports that fibrils containing more helical structures are larger in size [88]. The present findings indicated that the differences in the pathways of nucleation and/or fibril formations gave any perturbations to the angle β and the amount of helical contents in the fibrils, which would be tightly linked to the structural polymorphism of fibrils in the presence of NaCl.

## 4. Conclusions

In this study, we found that the amyloid fibrils of αS_57–102_ peptide on the membrane exhibited structural polymorphism with two morphologies (thin or thick fibrils) in the presence of NaCl but showed one morphology (thin fibrils) in the absence of NaCl. Further, the comprehensive analyses at the molecular level revealed that the presence of salt clearly depressed the membrane-bound region of αS_57–102_ and exposed the hydrophobic region to solvent, which induced two distinct pathways of fibril nucleation of αS_57–102_: one from the association of free αS_57–102_ with membrane-bound αS_57–102_ and the other from the assembly among membrane-bound αS_57–102_, resulting in the structural polymorphism. Despite some limitations in elucidating the fibrillation mechanism of full-length αS using the αS_57–102_ peptide (for example, the N- and C-terminal domains have lipid-binding properties and charge interactions, respectively, affecting the membrane affinity and fibril polymorphism of αS), the understanding of the fibrillization mechanism of αS_57–102_ would be helpful in elucidating the aggregation or fibrillization of full-length αS and the solvent-exposed hydrophobic regions identified in this study would be considered critical sites for the fibril nucleation because the NAC domain is established as the core region responsible for the β-sheet propensity and fibrillation of αS and its removal is known to significantly reduce aggregation, and the intermolecular interactions between NAC domains in αS_57–102_ directly promoted β-sheet formation and played a crucial role in nucleation. These findings would allow us to suggest that similar salt-driven polymorphism as observed in the αS_57–102_ fragment could also occur even for adding either the N- or C-terminal domain. Further, the exposed area might become potential therapeutic targets because strategies such as designing molecular compounds to interact with these regions or modulating intracellular salt concentrations could give some possibilities to control the aggregation processes, contributing to the therapeutic approaches. It is widely known that the morphology of fibrils formed by full-length α-synuclein varies depending on the type of salt and that fibril morphology is associated with toxicity. These findings also highlight the importance of investigating the effects of ions with different valences, such as ${\mathrm{Cu}}^{2+}$ and ${\mathrm{Zn}}^{2+}$, on fibril formation and their structural polymorphism.

## Figures and Tables

**Figure 1 biomolecules-15-00506-f001:**
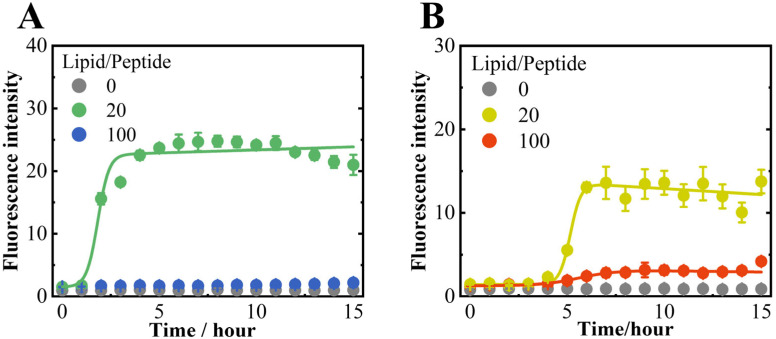
ThT fluorescence of αS_57–102_ at L/P ratios (lipid-to-peptide concentration ratios) of 0, 20, and 100 in the absence (**A**) or presence (**B**) of NaCl. Fluorescence measurements were performed three times per sample at 1-h intervals, and the averages were obtained. The excitation and emission wavelengths were 450 and 480 nm, respectively. The solid lines represent the fitted curves for each plot (see Section 2). The concentrations of ThT and αS_57–102_ were 10 and 50 μM, respectively, and the lipid membranes were composed of DMPG. Amyloid fibrils were formed by shaking the samples at 1500 rpm and 37 °C.

**Figure 2 biomolecules-15-00506-f002:**
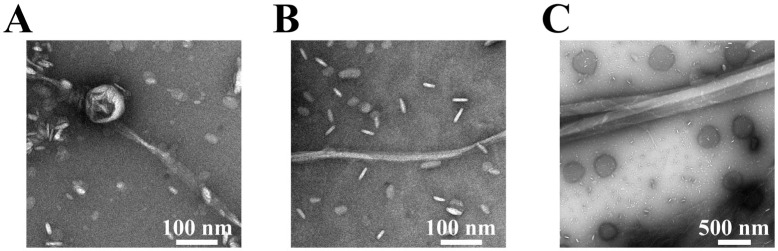
TEM images of amyloid fibrils at the L/P = 20 in the absence ((**A**), thin fibril; width = 18.6 ± 5.1 nm) or presence ((**B**), thin fibril; width = 13.2 ± 2.9 nm) and ((**C**), thick fibril; width = 156.2 ± 18.0 nm) of NaCl. The fibrils were prepared by incubating the ${\alpha S}_{57–102}$ at 1500 rpm and 37 °C for 10 h and adsorbing them onto carbon-coated copper grids, rinsing with water, and then negatively staining with 2% (*w*/*v*) uranyl acetate. The scale bars represent 100 nm in (**A**) and (**B**) and 500 nm in (**C**). (**A**,**B**) with 100 nm scale bars provide thin fibrils, while (**C**) with a 500 nm scale bar shows thick fibrils. The morphologies in (**B**,**C**) were observed in the same sample but at different scales.

**Figure 3 biomolecules-15-00506-f003:**
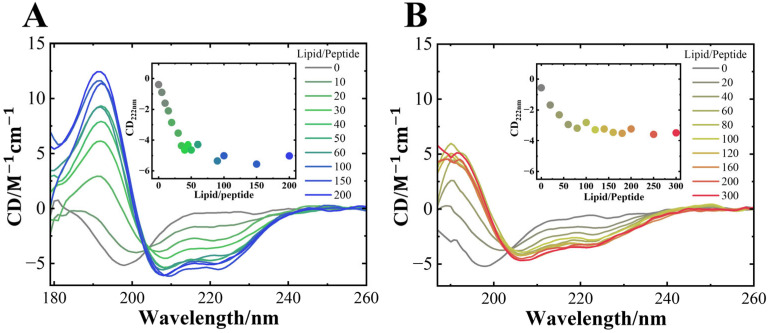
SRCD spectra of αS_57–102_ at various L/P ratios in the (**A**) absence and (**B**) presence of 0.1 M NaCl. The concentration of αS_57–102_ was 50 µM, and the lipid membranes were composed of DMPG. The spectra were acquired at a temperature of 25 °C, with an optical path length of 50 μm, a scanning speed of 20 nm/min, and eight accumulations. The inset shows the CD values at 222 nm plotted against the L/P ratios.

**Figure 4 biomolecules-15-00506-f004:**
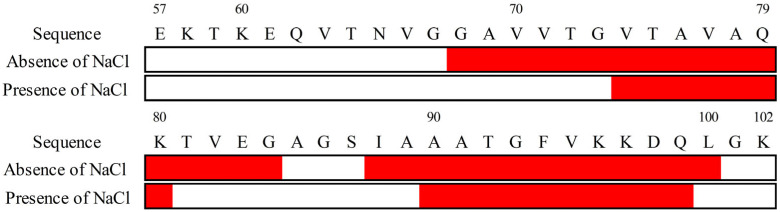
Sequences of secondary structures of membrane-bound αS_57–102_ in the absence or presence of NaCl (0.1 M). The red represents α-helix segments.

**Figure 5 biomolecules-15-00506-f005:**
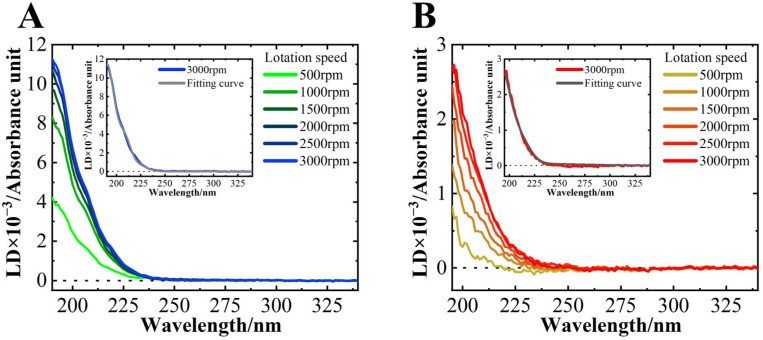
LD spectra of αS_57–102_ at various rotation speeds in the (**A**) absence or (**B**) presence of 0.1 M NaCl at an L/P = 200. The concentration of αS_57–102_ was 50 µM. The spectra were acquired at a temperature of 25 °C, with an optical path length of 500 μm, a data acquisition interval of 1 nm, a scanning speed of 20 nm/min, and two accumulations. The inset showed the fitting curve for the spectra at 3000 rpm.

**Figure 6 biomolecules-15-00506-f006:**
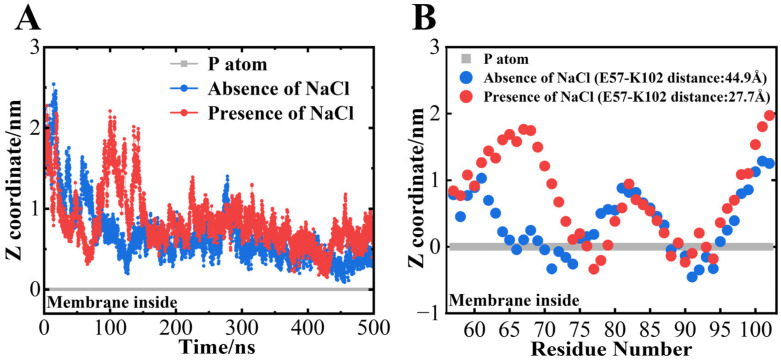
Interactions between αS_57–102_ and the DMPG membrane surface were characterized using an MD simulation. Time dependence of the distance between the center of gravity of (**A**) αS_57–102_ against the position of the phosphorus atom (set to 0) of the membrane surface. Average position of each amino acid residue of (**B**) αS_57–102_ (490–500 ns). Red and blue indicate the results in the presence and absence of NaCl, respectively.

**Figure 7 biomolecules-15-00506-f007:**
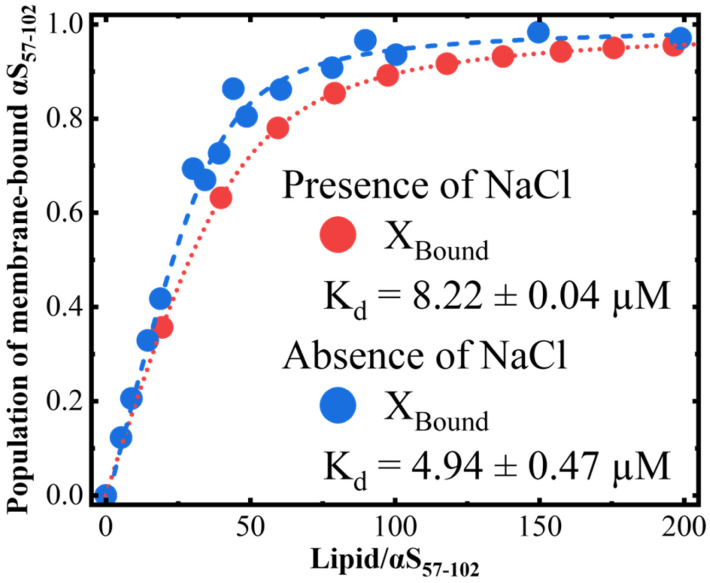
The population of membrane-bound αS_57–102_ was estimated from the L/P-dependent ${\mathrm{CD}}_{222\mathrm{nm}}$ values in the presence and absence of NaCl. $K_{d}$ represents the dissociation constant.

**Table 1 biomolecules-15-00506-t001:** Contents and numbers of segments of secondary structure estimated from the SRCD spectra of αS_57–102_ at the N and MB states in the presence or absence of NaCl.

	α-Helix		β-Strand		Turn	Unordered
	Contents (%)	Numbers	Contents (%)	Numbers	Contents (%)	Contents (%)
Native State (L/P = 0)						
Absence of NaCl	0	0	36.2 ± 1.7	4	16.9 ± 0.5	46.9 ± 0.7
Presence of NaCl	2.3 ± 0.5	1	29.1 ± 2.6	3	20.0 ± 1.2	48.6 ± 1.1
Membrane-bound State (L/P = 200)						
Absence of NaCl	66.4 ± 3.3	2	1.1 ± 2.5	1	20.6 ± 1.6	11.9 ± 3.0
Presence of NaCl	37.6 ± 1.5	2	12.8 ± 1.7	2	19.2 ± 1.6	30.4 ± 1.9

**Table 2 biomolecules-15-00506-t002:** Orientation of β-sheets within αS_57–102_ fibrils relative to the membrane normal (angle α) and the tilt angle of β-strands within the sheet plane (angle β) (Appendix A) calculated from ATR-FTIR data (Appendix A).

	Absence of NaCl	Presence of NaCl	Presence of NaCl
	L/P = 20	L/P = 20	L/P = 100
α-Helix contents (%)	28.7 ± 3.0	23.4 ± 5.9	33.8 ± 1.3
β-Sheet contents (%)	49.9 ± 0.4	55.5 ± 0.4	38.4 ± 1.0
Angle β (deg)	21.2 ± 5.3	33.2 ± 6.6	46.3 ± 0.9
Angle α (deg)	69.5 ± 2.2	67.2 ± 6.5	54.9 ± 2.5

## Data Availability

The data presented in this study are available on request from the corresponding author.

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
