# Peer review of "Salt-Induced Membrane-Bound Conformation of the NAC Domain of α-Synuclein Leads to Structural Polymorphism of Amyloid Fibrils"

_biomolecules, 2025, doi:10.3390/biom15040506_

Round 1
Reviewer 1 Report
Comments and Suggestions for Authors
This study provides critical insights into how sodium ions modulate the structural dynamics of the α-synuclein NAC domain (αS57–102) during membrane interactions and amyloid fibril formation. By integrating advanced biophysical techniques (synchrotron radiation circular dichroism, linear dichroism, and ATR-FTIR) with molecular dynamics simulations, the authors demonstrate that NaCl exposure induces conformational changes in the membrane-bound NAC domain. These changes promote two distinct fibrillization pathways—free peptide aggregation versus membrane-mediated assembly—resulting in structural polymorphism (thin vs. thick fibrils). The work advances our understanding of how ionic microenvironments in neuronal cells may influence Parkinson’s disease pathology by regulating fibril morphology.
While the focus on αS57–102 (NAC domain) provides valuable simplification for mechanistic studies, the exclusion of full-length α-synuclein leaves open whether the N- or C-terminal regions modulate salt sensitivity or fibril architecture. For example, the N-terminal domain’s lipid-binding properties or the C-terminal domain’s charge interactions might alter membrane affinity or aggregation kinetics. Could the authors comment on whether they anticipate similar salt-driven polymorphism in full-length α-synuclein, given its additional domains?
Overall, this work should advance understanding of environmental modulators in α-synuclein aggregation. While the use of a truncated peptide warrants discussion, the experimental rigor and novel findings justify acceptance. I recommend acceptance after a brief response to the question above in the revised manuscript.
Author Response
Answer to Reviewer 1
Thank you very much for the valuable comments and for giving us the opportunity to revise our manuscript. We have revised the manuscript according to your comments and suggestions as follows. We hope that these revisions are satisfactory and the revised version will be acceptable for publication in ‘Biomolecules’.
1-1) While the focus on αS57–102 (NAC domain) provides valuable simplification for mechanistic studies, the exclusion of full-length α-synuclein leaves open whether the N- or C-terminal regions modulate salt sensitivity or fibril architecture. For example, the N-terminal domain’s lipid-binding properties or the C-terminal domain’s charge interactions might alter membrane affinity or aggregation kinetics. Could the authors comment on whether they anticipate similar salt-driven polymorphism in full-length α-synuclein, given its additional domains?
Thank you for the valuable comments. As you pointed out, the N- and C-terminal domains of full-length αS play important roles in the membrane interaction processes [14] because the N- and C-terminal domains have lipid-binding properties and charge interactions, respectively, affecting the membrane affinity and fibril polymorphism. The αS57–102 fragment used in this study was selected because this region includes the NAC domain which is established as the core region responsible for the β-sheet propensity and fibrillation of αS [26] and its removal is known to significantly reduce aggregation. Furthermore, some studies using full-length αS have demonstrated that intermolecular interactions between NAC domains promoted β-sheet formation and played a crucial role in nucleation [25]. These findings would allow us to suggest that similar salt-driven polymorphism observed in αS57–102 fragment could be also occurred even for adding either the N- or C-terminal domain.
The related sentences were added in the section 4 (Conclusions) of revised manuscript (line 691–693, 696–698, and 699–702).
[14] Imaura, R.; Kawata, Y.; Matsuo, K. Salt-Induced Hydrophobic C-Terminal Region of α-Synuclein Triggers Its Fibrillation under the Mimic Physiologic Condition. Langmuir 2024, 40, 20537-20549, doi:10.1021/acs.langmuir.4c02178.
[25] Guzzo, A.; Delarue, P.; Rojas, A.; Nicolaï, A.; Maisuradze, G.G.; Senet, P. Wild-Type α-Synuclein and Variants Occur in Different Disordered Dimers and Pre-Fibrillar Conformations in Early Stage of Aggregation. Front. Mol. Biosci. 2022, 9, 910104.
[26] Sot, B.; Rubio-Muñoz, A.; Leal-Quintero, A.; Martínez-Sabando, J.; Marcilla, M.; Roodveldt, C.; Valpuesta, J.M. The chaperonin CCT inhibits assembly of α-synuclein amyloid fibrils by a specific, conformation-dependent interaction. Sci. Rep. 2017, 7, 40859.
Reviewer 2 Report
Comments and Suggestions for Authors
Imaura R and Matsuo K investigated salt-induced membrane-bound conformation of the NAC domain of α-synuclein. They found NaCl induced two morphologies of α-synuclein, thin and thick, while α-synuclein only forms the thin morphology in the absence of salts. Based on the conformation study, they proposed that αS57–102 has two distinct pathways of fibril nucleation, depending on the molar ratios of free and membrane-bound peptide, explaining the structural polymorphism. Here are my concerns that need to be clarified.
- In Figure 2, the statistical analysis for the thickness of fibrils is recommended.
- Does KCl have the same effect as NaCl?
- Page 15, Line 511, “αS5-102” should be “αS57–102”.
Author Response
Answer to Reviewer 2
Thank you very much for the valuable comments and for giving us the opportunity to revise our manuscript. We have revised the manuscript according to your comments and suggestions as follows. We hope that these revisions are satisfactory and the revised version will be acceptable for publication in ‘Biomolecules’.
2-1) In Figure 2, the statistical analysis for the thickness of fibrils is recommended.
Thank you for your adequate suggestion. We have conducted a statistical analysis for the thickness of fibrils using Image J software to clarify the differences in their morphologies, and found that the width of thin fibrils of L/P=20 was 18.6 ± 5.1 nm for NaCl absence and 13.2 ± 2.9 nm for NaCl presence, and that of thick fibrils in the presence of NaCl was 156.2 ± 18.0 nm for L/P=20 and 191.5 ± 19.5 nm for L/P=100. The related values were added in the Figure 2 and the section 3.1 of revised manuscript (line 380–381, 390).
2-2) Does KCl have the same effect as NaCl?
Thank you for your important comment. The effect of KCl for the structural polymorphism of fibrils was not conducted in our study. However, we expect that KCl does not induce the polymorphism as observed in NaCl presence because Havemeister et al. reported that NaCl accelerated fibril formation while KCl exhibited an inhibitory effect on fibril assembly [85]. Mechanistically, Na⁺ ions effectively neutralize electrostatic repulsion and stabilizes β-sheet-rich fibrillar structures, facilitating the nucleation and growth of fibrils and contributing to the formation of multiple polymorphic forms. Conversely, K⁺ ions lead to weaker reduction of electrostatic repulsion to maintain the monomeric protein with a random coil, inhibiting the formation of fibrils. Additionally, the fibrils formed in the presence of KCl increased its rigidity and decreased its clustering behavior, which made it difficult to form the fibril polymorphisms as observed in the presence of NaCl [85]. However, this study was conducted without lipid membranes and hence, further experimental validation under membrane-interacted conditions may be necessary to confirm the salt dependence of fibril polymorphisms.
These sentences were added in the section 3.5 of revised manuscript (line 635–648)
[85] Havemeister, F.; Ghaeidamini, M.; Esbjörner, E.K. Monovalent cations have different effects on the assembly kinetics and morphology of α-synuclein amyloid fibrils. Biochem. Biophys. Res. Commun. 2023, 679, 31-36, doi:10.1016/j.bbrc.2023.08.061.
2-3) Page 15, Line 511, “αS5-102” should be “αS57–102”
Thank you for your points. We corrected a typo in the section 3.4 of revised manuscript (line 511).
Round 2
Reviewer 2 Report
Comments and Suggestions for Authors
The authors addressed all my concerns.